# Physiological Adaptation Mechanisms to Drought and Rewatering in Water-Saving and Drought-Resistant Rice

**DOI:** 10.3390/ijms232214043

**Published:** 2022-11-14

**Authors:** Lele Wang, Xuenan Zhang, Yehong She, Chao Hu, Quan Wang, Liquan Wu, Cuicui You, Jian Ke, Haibing He

**Affiliations:** 1Agricultural College, Anhui Agricultural University, Hefei 230036, China; 2Jiangsu Collaborative Innovation Center for Modern Crop Production, Nanjing 210095, China

**Keywords:** rice, drought stress, photosynthetic physiology, antioxidant enzymes, osmoregulatory substances

## Abstract

Water-saving and drought-resistant rice (WDR) has high a yield potential in drought. However, the photosynthetic adaptation mechanisms of WDR to drought and rehydration have yet to be conclusively determined. Hanyou 73 (HY73, WDR) and Huanghuazhan (HHZ, drought-sensitive cultivar) rice cultivars were subjected to drought stress and rewatering when the soil water potential was −180 KPa in the booting stage. The leaf physiological characteristics were dynamically determined at 0 KPa, −30 KPa, −70 KPa, −180 KPa, the first, the fifth, and the tenth day after rewatering. It was found that the maximum net photosynthetic rate (*A*_max_) and light saturation point were decreased under drought conditions in both cultivars. The change in dark respiration rate (*R*_d_) in HY73 was not significant, but was markedly different in HHZ. After rewatering, the photosynthetic parameters of HY73 completely returned to the initial state, while the indices in HHZ did not recover. The antioxidant enzyme activities and osmoregulatory substance levels increased with worsening drought conditions and decreased with rewatering duration. HY73 had higher peroxidase (POD) activity as well as proline levels, and lower catalase (CAT) activity, ascorbate peroxidase (APX) activity, malondialdehyde (MDA) level, and soluble protein (SP) content during all of the assessment periods compared with HHZ. In addition, *A*_max_ was markedly negatively correlated with superoxide dismutase (SOD), POD, CAT, and SP in HY73 (*p* < 0.001), while in HHZ, it was negatively correlated with SOD, CAT, APX, MDA, Pro, and SP, and positively correlated with *R*_d_ (*p* < 0.001). These results suggest that WDR has a more simplified adaptation mechanism to protect photosynthetic apparatus from damage in drought and rehydration compared with drought-sensitive cultivars. The high POD activity and great SP content would be considered as important physiological bases to maintain high photosynthetic production potential in WDR.

## 1. Introduction

Due to global climate change, various climatic factors greatly impact agricultural production. Drought is one of the most serious abiotic stresses restricting agricultural development [1]. Rice is among the main food crops and the largest irrigated crop in agriculture; thus, shortages of water resources are bound to threaten its production [2,3]. At the same time, the sensitivity of rice to water at different growth stages is an important cause of drought stress [4,5].Drought stress leads to a decrease in rice photosynthesis rates and yields. Dry matter accumulation and yield gradually decrease with the aggravation of drought stress [6,7]. Therefore, it is of great significance to elucidate the drought resistance mechanisms of rice during the drought-sensitive period.

In recent years, water-saving and drought-resistant rice (WDR) varieties have been successfully cultivated. WDR, a new rice variety, is characterized by high yields, good quality, and the ability to save water and resist drought [8,9]. The high photosynthetic production potential and high-yield levels of WDR under drought can lead to high photosynthetic compensation effects after rewatering [10]. Physiological changes in rice leaves during drought stress have been comprehensively elucidated; however, the mechanisms involved in these physiological changes in WDR after rewatering have yet to be fully established. Therefore, it is of great significance to study the physiological effects of drought and rewatering on the leaves of WDR.

Antioxidant defense systems come into play when plants are under stress, scavenging reactive oxygen free radicals (ROS) and providing an intact cellular environment for the photosynthetic system [5,11,12,13]. Drought decreases photosynthesis and reduces the rate of electron transfer and the maximum quantum production of the crop, while different degrees and durations of drought cause different degrees of damage. All photosynthetic functions, especially the PSII system, can be recovered in sweet sorghum after drought rehydration, while the photosynthetic rate and photosynthetic electron transfer capacity of maize cannot be restored to pre-drought conditions under prolonged drought [14,15,16]. In addition, according to Zegada-Lizarazu and Monti (2013), photosystem II (PSII) photochemistry has an effective self-regulatory function for sweet sorghum, contributing to its high drought tolerance and photosynthetic resilience [14]. The inhibition of PSII activities under strong light is called photoinhibition [17,18]. Under drought stress, a large number of ROS are produced by photoinhibition and photorespiration [19]. The accumulation of ROS causes severe damage to the cell membrane system, resulting in the inactivation and hydrolysis of functional proteins [20,21]. There are enzymatic systems for scavenging ROS in plants, such as superoxide dismutase (SOD), peroxidase (POD), and catalase (CAT). SOD can scavenge O_2_^−^ in plants to form H_2_O_2_, which is then eliminated by POD and CAT [22,23]. As one of the most efficient enzymes, CAT plays an important role in decomposing H_2_O_2_ into O_2_ and H_2_O, while APX is responsible for the removal of H_2_O_2_ in the water–water and ASH–GSH cycles [20,24]. Under steady-state conditions, ROS molecules are cleared by various antioxidant defense machines [20]. However, the balance between ROS production and clearance is broken when plants are subjected to biotic or abiotic stress [25,26]. Under severe drought stress, a chain reaction can happen in plants with the balance of the antioxidant system broken and peroxide significantly increased. In addition, the occurrence of membrane lipid peroxidation destroys the cell membrane system to increase the membrane permeability.

MDA is the final product of membrane lipid peroxidation, which can reflect the degree of plant cell damage by stress and drought resistance of various varieties. Photoinhibition-mediated injury is majorly associated with reactive oxygen species production [27]. Excess light energy can transfer electrons to O_2_ on photosystem I (PSI) or PSII to make it O_2_^−^ [28]. Excess reactive oxygen species can lead to membrane lipid peroxidation, resulting in MDA accumulation and the destruction of the photosynthetic structure. MDA can also inhibit antioxidant enzyme activities or interact with protein and genetic material nucleic acids to deactivate them [29,30,31].

When water stress occurs, plant cells actively accumulate soluble osmotic regulatory substances such as proline and soluble proteins, reduce osmotic and water potentials, maintain turgor pressure, and carry out osmotic regulation [32,33]. These activities are aimed at maintaining the normal physiological processes of plant growth, stomatal movement, and photosynthesis [34]. However, the expression abilities of different osmotic regulatory substances may differ under corresponding drought stress. Currently, the photosynthetic protection and regulatory mechanisms of WDR are not clear.

After drought and rewatering, there is a need to elucidate the changes to the photosynthetic physiology, antioxidant enzyme activities, and osmotic regulation in WDR, and determine whether antioxidant enzymes and osmotic adjustment substances work in tandem to adapt to drought stress. We used the water-saving drought-resistant rice (HY73) and drought-sensitive rice (HHZ) varieties to study the changes in leaf physiological indices under drought stress and rewatering at the booting stage. This study aimed to: (1) reveal the dynamic characteristics of the photosynthetic production potential of WDR under drought and rewatering; (2) analyze the physiological basis of WDR’s response to drought and rewatering and its relationships with the photosynthetic production potential.

## 2. Results

### 2.1. Relative Water Content (RWC)

The relative water content (RWC) of the rice leaves decreased with increasing water stress, and after the rewatering of both rice varieties, there was a gradual increase in the initial state across both cultivars (Figure 1). From RW to ED, the RWC of HY73 and HHZ decreased by 20.32% and 29.33%, respectively. After rewatering, the RWC of HY73 returned to a steady state at the D1 stage, while HHZ was still slowly recovering. In general, HY73 had a higher RWC than HHZ from the HD to the D5 stage (Figure 1).

### 2.2. Photosynthetic Performance

Under different water potentials at the booting stage, the changing trends of the photosynthetic performance of the two rice varieties were the same, but the amplitudes were different (Figure 2). At the booting stage, *A*_max_ and *I*_s_ decreased with increasing drought stress levels and gradually increased after rewatering, while *I*_c_ exhibited the opposite trend. The *A*_max_ of HY73 decreased from 21.57 μmol·m^−2^·s^−1^ at the RW stage to 13.76 μmol·m^−2^·s^−1^ at the ED stage, while that of HHZ decreased from 29.82 μmol·m^−2^·s^−1^ at the RW stage to 11.71 μmol·m^−2^·s^−1^ at the ED stage. Changes in *A*_max_ of HY73 at the HD stage were not significant, while the decrease in *A*_max_ of HHZ was 34.74% when compared with *A*_max_ at the RW stage.

Regarding the *I*_c_ levels from RW to ED, those of HY 73 were increased from 16.71 μmol·m^−2^·s^−1^ to 38.89 μmol·m^−2^·s^−1^ with an increase of 132.73%, and decreased by 48.68% to 19.96 μmol·m^−2^·s^−1^ at the D1 stage when compared with the ED stage. For cultivar HHZ, the *I*_c_ levels were 15.87 μmol·m^−2^·s^−1^ and 30.20 μmol·m^−2^·s^−1^ at the RW and ED stages, respectively, and then decreased to 22.03 μmol·m^−2^·s^−1^ at a rate of 27.05%, 15.53%, and 3.0%, respectively, at the D1, D5, and D10 stages in cultivar HHZ. The *I*_s_ levels in cultivar HY73 decreased from 1825.88 μmol·m^−2^·s^−1^ (RW) to 1283.71 μmol·m^−2^·s^−1^ (ED), while those of HHZ decreased from 1649.49 μmol·m^−2^·s^−1^ (RW) to 1110.74 μmol·m^−2^·s^−1^ (ED). At the D1 stage, the *I*_s_ of HHZ were still gradually decreasing, while those of HY73 had recovered significantly to 1586.48 μmol·m^−2^·s^−1^. At D5 and D10, the *I*_s_ of HHZ increased to 1325.30 μmol·m^−2^·s^−1^ and 1503.73 μmol·m^−2^·s^−1^, respectively.

The amplitude of the change of *R*_d_ was small in cultivar HY 73, but was greater in HHZ. The highest and lowest *R*_d_ of HHZ were at RW (2.52 μmol·m^−2^·s^−1^) and D1 (0.96 μmol·m^−2^·s^−1^), respectively.

### 2.3. Antioxidant Enzyme Activities

The SOD activities in the two varieties were markedly increased under drought and significantly decreased after rewatering (Figure 3). The SOD activities of HY73 and HHZ at ED were increased by 68.86% and 135.37%, respectively, and respectively decreased by 40.99% and 45.46% at the D10 stage when compared with the RW stage. The activities of POD, CAT, and APX were significantly increased under drought, in which the parameters exhibited significant differences between the HD and ED stages (*p* ≤ 0.05). All of the above-mentioned parameters declined after rewatering. At the HD stage, the POD, CAT, and APX activities of HY73 increased by 35.90%, 21.48%, and 75.12%, respectively, while those of HHZ increased by 28.56%, 31.64%, and 51.25%, respectively, compared with the RW stage. At the ED stage, the POD, CAT, and APX activities of HY73 were respectively increased by 6.61%, 25.71%, and 52.09%, and in HHZ were reduced by 9.34% and increased by 38.60% and 26.81%, respectively, compared with the HD stage. At the D10 stage, the POD, CAT, and APX activities of HY 73 respectively decreased by 25.09%, 50.24%, and 51.95%, while those of HHZ decreased by 20.95%, 38.45%, and 32.61%, respectively, compared with the ED stage. In general, the activities of the antioxidant enzymes in the leaves of HY73 quickly responded under drought and decreased to the initial value after rewatering. However, the activities of SOD and CAT in the HHZ leaves reached their highest levels at ED, while the POD levels were at the highest level at the HD stage. There was always a certain gap between rehydration and the initial state as the initial state could not be restored.

### 2.4. MDA Levels

The MDA levels in the rice leaves were markedly increased with increasing drought at the booting stage and rapidly decreased to stable levels after rewatering. There were significant differences in the leaf MDA levels between the two rice varieties under each observed stage. The MDA content of HY73 showed a 45.88% increase, 4.95% decrease, 23.05% decrease, and 24.15% decrease in HD, ED, D1, and D10, respectively, while in HHZ, it was a 45.74% increase, 63.99% increase, 31.90% decrease, and 6.80% decrease, respectively (Figure 4).

### 2.5. Osmotic Adjustment Substance Content

The osmotic adjustment substance levels of the WDR and drought-sensitive rice varieties exhibited the same change trend under different water potentials, which increased significantly with worsening drought stress and decreased with the extension of rewatering time. Under drought stress, the Pro accumulation levels and rates in HY73 were higher than those in HHZ; after rewatering, the Pro levels in HY73 rapidly decreased compared to those of HHZ. The changes in Pro levels in HY73 were 140.79% increased, 29.52% increased, 30.60% decreased, 28.73% decreased, and 41.08% decreased in HD, ED, D1, D5, and D10, respectively, while the Pro levels in HHZ were 40.09% increased, 42.12% increased, 9.16% decreased, 9.58% decreased, and 6.19% decreased, respectively. The Pro levels in HD and ED were 1.95 and 1.78 times higher in HY73 than HHZ, respectively. Under different water conditions, the SP levels of HY73 were 28.36% greater, 33.54% greater, 32.10% lower, 19.42% lower, and 6.63% lower, while the SP levels of HHZ were 34.14% greater, 19.95% greater, 8.50% lower, 6.91% lower, and 9.81% lower in HD, ED, D1, D5, and D10, respectively (Figure 5).

### 2.6. Correlation Analysis

The *A*_max_ of HY73 was negatively correlated with the levels of SOD, CAT, and SP (*p* < 0.001, Figure 6), while the *A*_max_ of HHZ was negatively correlated with the activities of CAT, APX, MDA, Pro, and SP, and positively correlated with *R*_d_ (*p* < 0.001, Figure 7).

## 3. Discussion

### 3.1. Photosynthetic Performance in Drought and Rewatering of WDR

Drought stress can lead to a decrease in relative water levels, *A*_max_, *I*_s_, and *R*_d_, as well as an increase in *I*_c_ (Figure 2). Under stress conditions, the rate of photosynthesis decreases, the photosynthetic capacity is reduced, and the demand for electrons in photosystem reaction centers are lowered. However, electron transport continues, leading to the transfer of excess electrons from PSII to PSI and thus the production of reactive oxygen free radicals that damage the PSI [35,36]. Studies have reported on the sensitivity of PSII to photodamage, but little is known about the photodamage of PSI [37]. SOD can scavenge the ROS generated by the reducing side of PSI, but when SOD is inactivated, it can no longer protect PSI from reactive oxygen species-mediated damage [38,39,40]. Since the damage to PSI is irreversible, it may have stronger effects on leaf survival outcomes than PSII photoinhibition. PSI is rarely damaged, but always recovers slowly and incompletely after damage and degradation [41,42,43]. In this study, the *A*_max_ of HHZ could not return to the state of RW after rehydration, which may be due to a damaged PSI system. High *R*_d_ increases organic matter consumption and results in a decrease in *A*_max_ and dry matter accumulation, and a final decrease in yield [44]. In this study, there were no marked changes in the *R*_d_ of HY73, which was markedly decreased in HHZ with worsening drought. These outcomes could have resulted in the reduction of energy consumption and yield loss in HHZ [45,46].

According to the established methods for monitoring light damage and in vivo repair of PSII in plants, ROS mainly inhibit the repair of PSII rather than directly destroying PSII [47,48,49], while the inhibition of ROS on the photodamage repair of PSII is reversible [50]. The light saturation point and light compensation point are important indices that reflect the utilization ability of strong and weak light by plants [51,52,53]. The *I*_s_ and *I*_c_ in the rice leaves respectively decreased and increased with worsening drought stress, indicating that drought stress reduced the utilization ability of the rice plants to high and low light intensity at the same time [51,52,53]. In general, under drought and rewatering, HY73 has a higher *I*_s_ and lower *I*_c_ than HHZ (Figure 2), indicating that HY73 may have a wider range of photo intensity utilization than HHZ. The *A*_max_ of HY73 was significantly negatively correlated with *I*_c_, positively correlated with *I*_s_, but not significantly correlated with *R*_d_ (Figure 6), while the *A*_max_ of HHZ was not significantly correlated with *I*_c_ or *I*_s_, but positively correlated with *R*_d_ (Figure 7). Differences in the correlations between *A*_max_ and other photosynthetic parameters may be the reason why WDR maintains its high photosynthetic potential under drought stress.

### 3.2. Effects of Antioxidant Enzyme Activities on Photosynthetic Production in Drought and Rewatering of WDR

To adapt, drought stress leads to changes in the ecological physiology of rice [54,55,56]. The electron spin limitation of O_2_^-^ enhances its acceptability of electrons, resulting in ROS production. Oxidative stress is a kind of damage to reactive oxygen free radicals caused by water stress, which causes damage to plants by producing reactive oxygen species. The scavenging of reactive oxygen species requires antioxidant enzymes such as SOD, POD, and CAT, among others [57,58,59]. Under water stress, the activities of antioxidant enzymes in the rice leaves increased across the cultivars (Figure 3). Our results are consistent with previous studies [41,58]. Meanwhile, there are significant differences in enzymatic activities among the different genotypic varieties. In general, the greater the increase in enzyme activities in varieties with strong drought resistance, the stronger their ability to resist stress [58,59]. Our results were in agreement with those of previous studies (Figure 3). In this study, under drought stress and rehydration, there was no obvious difference in either cultivar during the drought and rehydration processes for the SOD parameter. For POD activity, HY73 had higher activity, especially during drought, to firstly eliminate ROS substances when compared with HHZ. However, cultivar HHZ activated the CAT and APX approaches to further resist damage from ROS by improving both of the enzymes’ activities because of poor protection capabilities from the POD approach compared with HY73 (Figure 3). More drought-associated defense mechanisms can lead to more energy loss and a reduction of net photosynthetic capacities. In addition, there was a significant relationship between *A*_max_ and POD activity in HY73 (Figure 6; *p* < 0.001). No significant correlational relationship was observed between *A*_max_ and POD activity in HHZ (Figure 7; *p* > 0.05). These results indicate that HY73 has sufficient capacities to clear reactive oxygen species by enhancing POD activities. However, the drought-sensitive variety (HHZ) needs to activate more defense mechanisms such as CAT and APX approaches in addition to the SOD and POD pathways (Figure 3).

At a certain range, the scavenging abilities of reactive oxygen free radicals in rice increased with increasing degree of stress, and beyond a certain range, the scavenging abilities of the reactive oxygen free radicals in the rice decreased, resulting in accelerated leaf senescence and increased MDA levels [60,61,62,63,64]. Compared with the varieties with weak drought resistance abilities, the varieties with strong drought resistance abilities exhibited a smaller increase in MDA levels. Varieties with strong drought resistance abilities have higher reactive oxygen species scavenging abilities to eliminate excess MDA and reduce MDA levels (Figure 4) [65]. In this study, under drought treatment, there were marked changes in the MDA levels of the rice, consistent with findings from previous studies, which reported that drought increased MDA levels, damaged the cell membrane, and was negatively correlated with the drought resistance of plants [66,67,68]. In this study, the MDA levels in HHZ were higher, which may be due to elevated levels of MDA in the cell membrane. Previous studies showed that there were some differences in changes in the MDA levels among varieties [58,59,65]. Compared with the varieties with weak drought resistance abilities, varieties with strong drought resistance abilities usually exhibit small increases in MDA levels, which supports our findings. The activities of the endogenous reactive oxygen scavenging enzyme of POD was higher in HY73. Thus, improving POD activity would be an important self-protective mechanism to maintain high photosynthetic potentials when WDR responds to drought.

### 3.3. Effects of Osmoregulatory Substances on Photosynthetic Production in Drought and Rewatering of WDR

Osmoregulatory substances play an important role in regulating cell osmotic potential and plant resistance to water stress [69,70]. Under water stress, Pro is the most effective osmoregulatory substance in plants and is closely associated with plant drought resistance [55,56,57]. The accumulation of free Pro in rice depends on rice water changes and is also related to the existence of a drought-resistance mechanism in rice [58,59,60,61]. In this study, the accumulation of free Pro in the rice increased with increasing treatment time (Figure 5). The Pro and SP levels increased with increasing drought degree, which mediated the correlation between the production of ROS (mainly hydroxyl radical) and the scavenging of ROS [71,72,73]. After drought and rehydration, there were consistent trends in the SP and Pro levels, indicating that the two had synergistic effects on cellular osmoregulation under drought stress. SP is the main product of plant photosynthesis and a nonstructural carbohydrate [74]. Changes in the SP content under drought stress can not only reveal its osmoregulatory capacity, but can also be used to probe the photosynthetic assimilation capacity of the plant itself. The correlations between the *A*_max_ of HY73 and SP and Pro showed differences, in a sense verifying this point. Under drought stress, the Pro levels in both varieties were positively correlated with the degree of drought. After rehydration, the Pro levels of HY73 rapidly decreased, relative to those of HHZ, indicating that HY73 can osmotically condition more rapidly than HHZ, which is important for reducing membrane lipid damage.

## 4. Materials and Methods

### 4.1. Experimental Design

This study was conducted from May to November 2021 at Wanzhong Comprehensive Experimental Station, Lujiang County, Hefei City, Anhui Province (31°48′ N, 117°23′ E). The pots had a volume of 0.026 m^3^ with a diameter of 30 cm and a height of 40 cm. The soil used to fill the pots was perennial paddy soil (a typical sandy loam) that had been harvested from the 0–20 cm layer at the experimental station. The physical and chemical properties of the soil were: organic matter levels of 32.40 mg·kg^−1^, total N levels of 2.00 g·kg^−1^, organic phosphorus levels of 24.80 mg·kg^−1^, total K levels of 19.40 g·kg^−1^, pH of 5.9, and soil bulk level of 1.15 g·cm^−3^. Each barrel contained 20 kg of fine-sifted soil.

The rice varieties were HY73 and HHZ, of which HY73 is a three-line hybrid rice with high-yield, high-quality, water-saving, and drought-resistant properties, while HHZ is an indica conventional weakly drought-resistant rice variety. Drought stress treatments were performed during the gestation period, and they included regular watering (RW, 0 KPa), moderate drought stress (MD, −30 KPa), severe drought stress (HD, −70 KPa), and extreme drought stress (ED, −180 KPa), with rehydration one day after ED and the same water management before water control until the end of the physiological period. The relevant indices were measured and collected on the first (D1), fifth (D5) and tenth (D10) days after rehydration. Each process was repeated three times. Maintain a 2–3 cm layer of water before water control, remove water from buckets in the evening before the day water control, and stop watering after the start of stress. Soil water potential is measured twice daily depending on weather conditions, rehydrated when it approaches or exceeds the bottom limit of the stress level, and remains at this drought stress level. The soil water potential was monitored using a tensiometer (Watermark, Irrometer Company Riverside, Riverside, CA, USA).

We used the completely randomized experimental study design. Twenty-one-day-old seedlings were removed from the seedling tray and transplanted into each pot. Each pot was supplemented with 2.4 g N, 1.2 g P_2_O_5_, and 3 g K_2_O. Moreover, they were top-dressed with 30% N, 50% K_2_O, and P_2_O_5_. The residual N rate was applied at the early tiller stage (30% of N application) and panicle differentiation stage (40% of N application). In addition, 50% K_2_O fertilizer was applied at the panicle differentiation stage.

### 4.2. Measurements

#### 4.2.1. Relative Water Levels

Three replicates of six to eight leaves per treatment were obtained. The fresh leaf weights (m_1_) were immediately taken after the leaves had been sliced from the base, after which the leaves for each treatment were cut into small sections, soaked in distilled water for 12 h, and weighed to obtain the saturated leaf weight (m_2_). Then, they were placed into the oven for 2 h at 105 °C, baked at 85 °C to a constant weight, and weighed to obtain the dry weight (m_3_). The formula for determining the relative water content of leaves W(%) was: (m_1_ − m_3_)/(m_2_ − m_3_) × 100%.

#### 4.2.2. Photosynthetic Light-Response Curve

A Li-6400 (Li-CorInc, USA) portable photosynthesis meter was used with its own red and blue light source to provide a constant light intensity of 1500 μmol·m^−2^·s^−1^, using a CO_2_ cylinder to control the concentration of CO_2_ to 400 μmol·m^−2^·s^−1^. The maximal PAR setting in the experiment was determined based on the range of light intensities during normal daylight from 09:00 to 12:00 at the test site last year, for the purpose of rapid light acclimatization of the leaves and to avoid phenomena such as photoinhibition. In the greenhouse, the temperature was controlled at 25 ± 2.3 °C, while the humidity was controlled at 68 ± 4.8%. The representative fully unfolded flag leaves were selected and the widest part of the leaves measured. Light-response curves for each variety were measured with three independent biological replicates. The maximum waiting time was 300 s, while the minimum waiting time was 180 s. The light-response curves were set at 1500, 1200, 800, 500, 300, 150, 100, 50, 25, and 0 μmol·m^−2^·s^−1^. After determining the light-response curve, the leaves were first treated with liquid nitrogen and then stored in the −80 °C cryogenic refrigerator for the determination of antioxidant enzyme activities and other physiological indices. The light-response curve was fitted by the light response curve model under arbitrary light intensity, as reported by Ye et al. (2007) [75]. The model expression is:(1)$$A_{n}\left(I\right)=\alpha \frac{1-\beta I}{1+\gamma I}\left(I-I_{c}\right),$$
if *I* = 0,
(2)$$A_{n}\;\left(0\right)=R_{d}=-\alpha I_{c},$$

The light saturation point *I*_s_ was obtained using the Formula (1),
(3)$$I_{s}=\frac{\sqrt{\left(\beta +\gamma \right)1+\gamma Ic/\beta }-1}{\gamma },$$and max photosynthesis rate *A*_max_ was obtained when *I* = *I*_s_,
(4)$$A_{\max}=A\left(Is\right)=\alpha \frac{1-\beta Is}{1+\gamma Is}Is-Ic,$$

In the formula, *A*_n_ (*I*) represents the net photosynthetic rate when the light intensity is *I*; *α*, *β*, and *γ* are the parameters independent of light intensity; *I*_c_ represents the light compensation point; and *R*_d_ represents the dark respiration rate.

#### 4.2.3. Determination of Antioxidant Enzyme Activity Levels (SOD, POD, CAT)

In this assay, 0.5 g of fresh sample was mixed with 5 mL of pre-chilled 50 mmol·L^−1^ phosphate buffer (pH 7.0), ground on ice, the volume increased to 10 mL, centrifuged at 4 °C, 13,000 r·min^−1^ for 20 min, and the supernatant obtained as the crude enzyme extract. The crude enzyme extract was used for the determination of SOD, POD, CAT enzyme activities, and soluble protein levels. Superoxide dismutase (SOD) activities were determined using the nitrogen blue tetrazolium (NBT) method; catalase (CAT) activities were determined using the UV spectrophotometric method; peroxidase (POD) activities were determined by the guaiacol method.

#### 4.2.4. Determination of APX Enzyme Activities

In this assay, 1 g of leaf tissue with main veins removed was weighed and mixed with 1.6 mL of pre-cooled phosphate buffer (pH 7.8) extract (containing 1 mmol·L^−1^ AsA, 3 mmol·L^−1^ *β*-mercaptoethanol, 0.5 mmol·L^−1^ PMSF, 2% PVP, 1 mM EDTA). The extracts were ground with liquid nitrogen, centrifuged at 4 °C, 13,000 r·min^−1^ for 20 min, and the supernatant used for determination of enzyme activities. About 0.10 mL of the enzyme solution was mixed with 1.70 mL of PBS containing 0.1 mM EDTA-Na_2_ (0.05 mol·L^−1^, pH 7.0), 0.10 mL of 5 mM AsA, and finally with 0.10 mL of 20 mM H_2_O_2_. The optical densities were determined at OD290 and the enzyme activities determined.

#### 4.2.5. Analyses of Osmoregulatory Compounds (Malondialdehyde, Proline, Soluble Protein)

The malondialdehyde level determination was performed using the thiobarbituric acid method with about 0.5 g of cut and mixed rice leaves with the main veins removed in a mortar. The samples were mixed with a small amount of quartz sand and ground rapidly with liquid nitrogen to powder. Then, 2.5 mL of trichloroacetic acid (TCA) was added to the samples, ground into a homogenate, supplemented with 7.5 mL of TCA, and ground further. Then, they were transferred into test tubes, centrifuged at 4000× *g* for 10 min, and 2 mL of supernatants were obtained. The supernatants were mixed with 2 mL of 0.6% thiobarbituric acid (TBA), placed in a boiling water bath for 15 min, rapidly cooled, and centrifuged at 4000× *g* for 15 min. Then, the absorbance values of the supernatants were determined at 532 nm, 600 nm, and 450 nm. The results were calculated as: concentration of MDA (μmol·mol^−1^) = 6.45 × (OD532 − OD600) − 0.56OD450.

Determination of proline levels: In this assay, to 5 mL of 3% sulfosalicylic acid, a plug was added, boiled in a water bath for 10 min, 2 mL of extract solution was aspirated, and 2 mL of glacial acetic acid and 3 mL of acid ninhydrin were added, and boiled in a water bath for 1 h. After cooling, 4 mL of toluene was added to each tube, shaken for 30 s, left to stand for a while, and the upper red proline layer was gently aspirated for measurement of absorbance at 520 nm wavelength.

Determination of soluble protein levels: here, 20 μL of extract solution (enzyme solution) was mixed with 80 μL, pH 7.8 phosphate buffer (i.e., diluted to 0.1 mL extract solution), supplemented with 2.9 mL of Thomas Blue solution, and the optical density measured at OD595 after 2 min of reaction (0.1 mL of buffer solution plus 2.9 mL of Thomas Blue was used as the control for zeroing).

### 4.3. Statistical Analysis

The test results are expressed as the average value and standard error of three replicate analyses. Differences between the averages of the parameters were compared by Fisher’s Least Significant Difference (LSD) test at the 5% level of significance in SPSS 26.0. Light response curve models for *A*_max_, *R*_d_, *I*_c_, and *I*_s_, Pearson correlation analysis for correlation plots, as well as the figures were all operated by OriginPro2021.

## 5. Conclusions

(i) Drought stress decreases photosynthetic rates and generally increase the activities of antioxidant enzymes and the content of osmoregulatory substances in both rice cultivars. WDR (HY73) can be rapidly recovered to its initial pre-drought state after rehydration, while the photosynthetic potential is unable to recover after rehydration for the drought-sensitive cultivar (HHZ).

(ii) High photosynthetic potentials mainly rely on high POD activity to eliminate reactive oxygen species for the protection of cell membrane integrity in WDR. In addition, the high SP content in WDR has a positive role in maintaining a steady intracellular osmotic pressure. Both factors could be essential physiological bases for the rapid recovery of photosynthetic potential after rehydration in WDR.

## Figures and Tables

**Figure 1 ijms-23-14043-f001:**
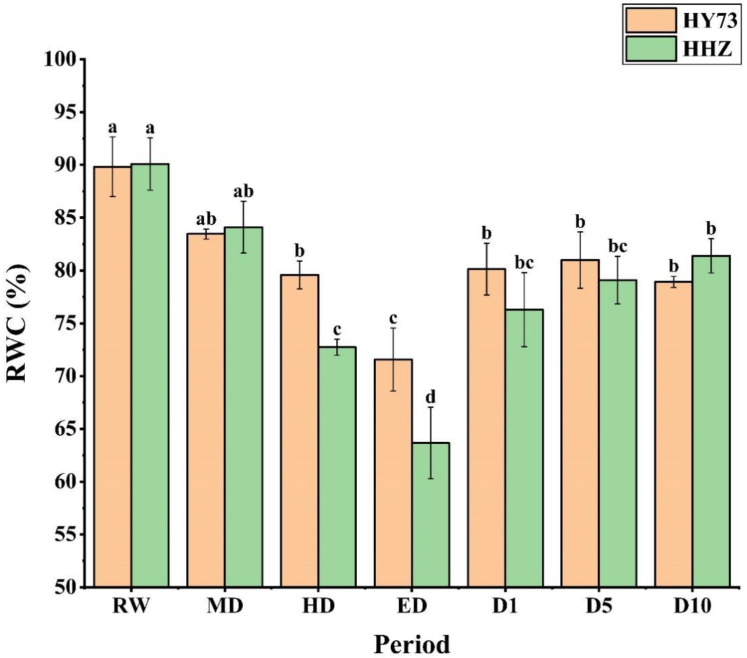
Relative water content in rice leaves under different water conditions at the booting stage: RW: regular watering; MD: moderate drought; HD: severe drought; ED: extreme drought; D1, D5, and D10: first, fifth, and tenth day after rehydration, respectively. Different letters indicate significant differences in variables among various treatments according to LSD (*p* ≤ 0.05), and vertical bars represent standard errors. Three biological replicates per index.

**Figure 2 ijms-23-14043-f002:**
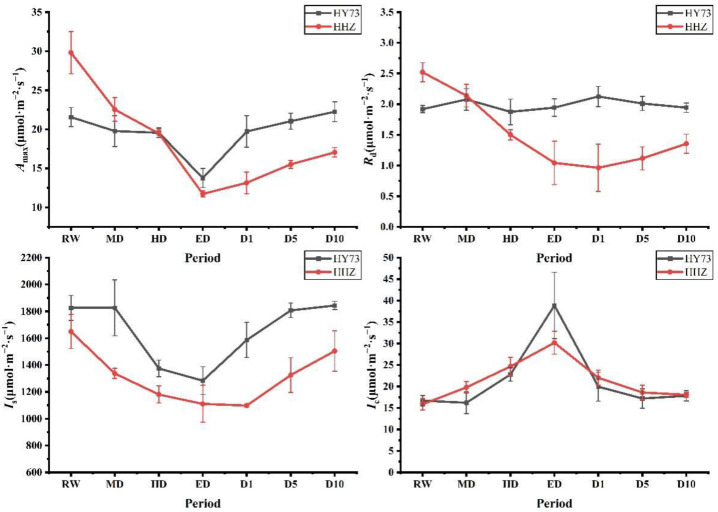
Dynamic characteristics of photosynthetic performance after drought and rewatering at the booting stage. RW: regular watering; MD: moderate drought; HD: severe drought; ED: extreme drought; D1: the first day after rehydration; D5: the fifth day after rehydration; D10: the tenth day after rehydration. Different letters indicate significant differences in variables among various treatments according to LSD (*p* ≤ 0.05), and vertical bars represent standard errors. Three biological replicates per index.

**Figure 3 ijms-23-14043-f003:**
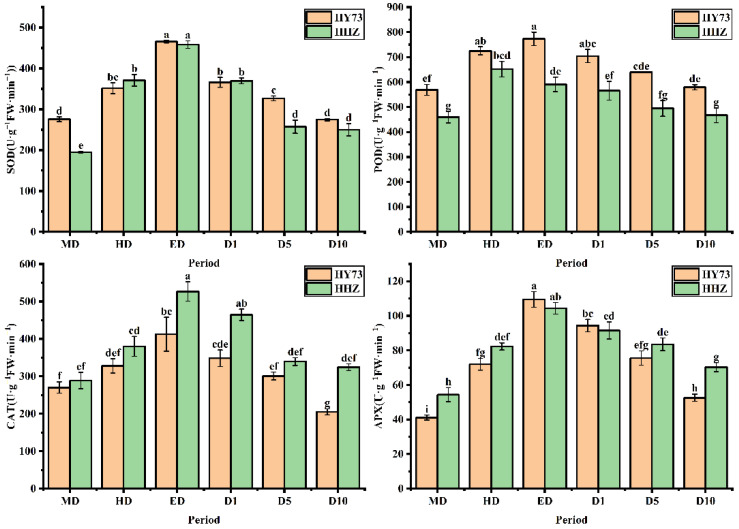
Activities of antioxidant enzymes in leaves of rice under different water conditions at the booting stage. MD: moderate drought; HD: severe drought; ED: extreme drought; D1: the first day after rehydration; D5: the fifth day after rehydration; D10: the tenth day after rehydration. Different letters indicate significant differences in variables among various treatments according to LSD (*p* ≤ 0.05), and vertical bars represent standard errors. Three biological replicates per index.

**Figure 4 ijms-23-14043-f004:**
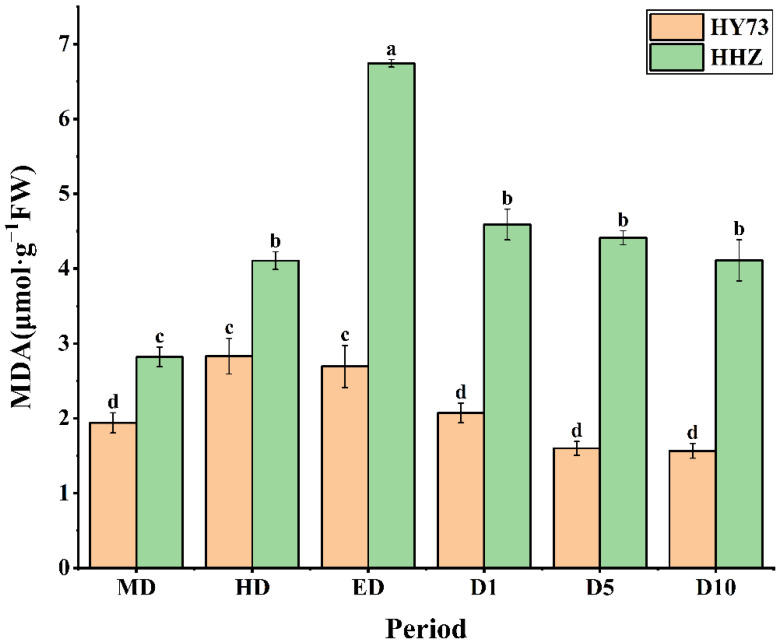
MDA content in rice under different water conditions at the booting stage. MD: moderate drought; HD: severe drought; ED: extreme drought; D1: the first day after rehydration; D5: the fifth day after rehydration; D10: the tenth day after rehydration. Different letters indicate significant differences in variables among various treatments according to LSD (*p* ≤ 0.05), and vertical bars represent standard errors. Three biological replicates per index.

**Figure 5 ijms-23-14043-f005:**
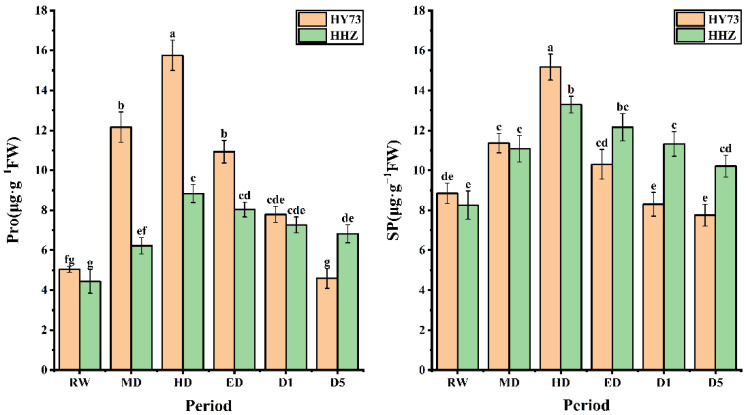
MDA content in rice under different water conditions at the booting stage. RW: regular watering; MD: moderate drought; HD: severe drought; ED: extreme drought; D1: the first day after rehydration; D5: the fifth day after rehydration; D10: the tenth day after rehydration. Different letters indicate significant differences in variables among various treatments according to LSD (*p* ≤ 0.05), and vertical bars represent standard errors. Three biological replicates per index.

**Figure 6 ijms-23-14043-f006:**
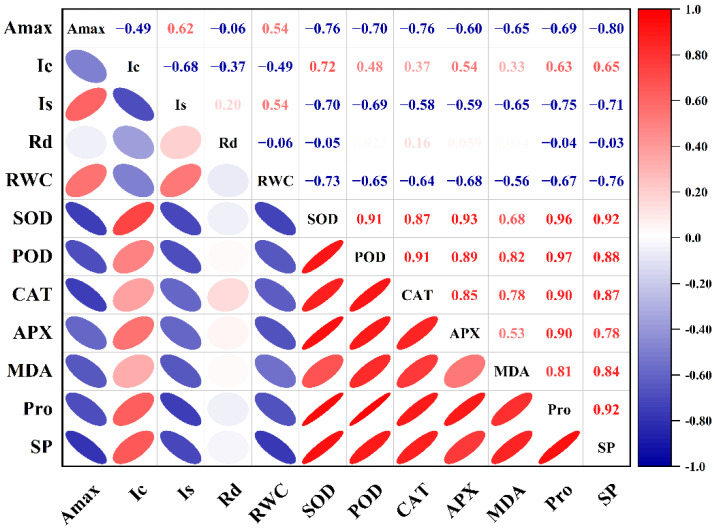
Correlation between photosynthetic parameters and physiological parameters of HY73. The data from cultivars and water treatments were merged to calculate the correlation coefficients (n = 18).

**Figure 7 ijms-23-14043-f007:**
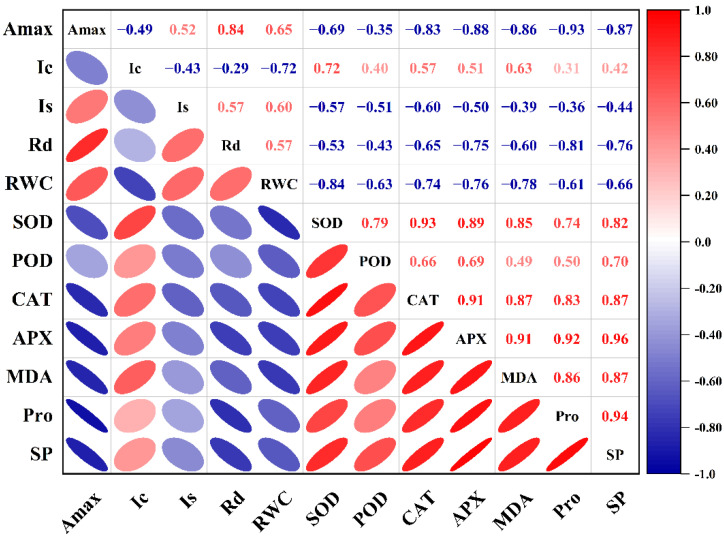
Correlation between photosynthetic parameters and physiological parameters of HHZ. The data from cultivars and water treatments were merged to calculate the correlation coefficients (n = 18).

## Data Availability

Not applicable.

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
