# Peer review of "Physiological Adaptation Mechanisms to Drought and Rewatering in Water-Saving and Drought-Resistant Rice"

_ijms, 2022, doi:10.3390/ijms232214043_

Round 1

Author Response

Response to Reviewer #1

Comment 1: Line 18, Abbreviation for Pmax is not appropriate for the first time.

Response 1: Thanks to your guidance, the abbreviations that appear for the first time in the text have been re-added in their full spelling form, Pmax in the text refers to ‘maximum net photosynthetic rate’. Whereas Amax is more commonly used to represent this index, so Pmax is replaced by Amax throughout the text. 

Comment 2: The treat time of drought stress should be clear in the abstract.

Response 2: Thanks a lot. We gladly accept your suggestion. In the abstract, the point that the drought treatment period is really the gestation period is not explicitly shown. It has now been added.

Comment 3: Line27 to 28, it would be better if clearer conclusion presented here.

Response 3: Thank you. We accepted your friendly suggestion. Now, the sentence was corrected. Please see lines 30 to 33.

These results suggest that WDR has a simplified adaptation mechanism to protect photosynthetic apparatus from damage in drought and rehydration compared with drought sensitive cultivar. The high POD activity and great SP content would be considered as important physiological bases to maintain high photosynthetic production potential in WDR.

Comment 4: In introduction, it would be necessary to introduce how the photosynthetic apparatus of plants response to drought and re-watering.

Response 4: Grateful for your suggestion, the introduction did lack a description of the photosynthetic apparatus in response to drought and rehydration and this has now been added as follows.Please see lines 58 to 66.

Drought decreases photosynthesis and reduces the rate of electron transfer and the maximum quantum production of the crop, while different degrees and durations of drought cause different degrees of damage. All photosynthetic functions, especially the PSII system, can be recovered in sweet sor-ghum after drought rehydration, while the photosynthetic rate and photosynthetic electron transfer capacity of maize cannot be restored to pre-drought conditions under prolonged drought[14–16]. In addition, according to Zegada-Lizarazu&Monti (2013), photosystem II (PSII) photochemistry has an effective self-regulatory function for sweet sorghum, contributing to its high drought tolerance and photosynthetic resilience[14].

Comment 5: Lines 93 to 94, this study aims are general. Please re-wrote the sentence with more details.

Response 5: Much obliged to you for your suggestion, we have rewritten the study objectives with more details as below. Please see lines 103 to 106 

This study aimed at: (1). Revealing the dynamics characteristics of photosynthetic production potential of WDR under drought and re-watering. (2). Analyzing the phys-iological basis of WDR response to drought and re-watering and its relationships with photosynthetic production potential.

Comment 6: Line305, 308 add the blank space.

Response 6: Thank you, we have fixed the mistake.

Comment 7: Line306 delete the words “were the same”.

Response 7: Thank you for your correction, it has been removed.

Comment 8:line309, Not detailed description. It is recommended that to describe the typical characteristics of the WDR.

Response 8: Thankfully, a detailed description of the typical characteristics of a WDR has been added, as follows. Please see line 338 to 339.

The rice varieties were HY73 and HHZ, of which HY73 is a three-line hybrid rice with high yield, high quality, saving water, and drought resistance, and HHZ is an indica conventional weakly drought-resistant rice variety.

Comment 9: Original data or fitted equations of Pn-PAR curves should be presented in text. Regrettably, i have not seen the information except for the derivative parameters (Fig. 2) deduced from light response curve model.

Response 9: Thank you for your comments, the fitted equations for the Pn-PAR curves have been provided as supplementary data.

Comment 10: The figures are not blight and seem to be low resolution. Please provide high resolution figures, especially for Figures 2, 3, and 5.

Response 10: Thanks to your advice, the high-resolution versions of Figures 2, 3, and 5 have been updated in the text.

Comment 11: Methods: lines 315 to 317, line 338, please re-wrote the sentence.

Response 11: Thank you. Lines 315 to 317, line 338 respectively have now been changed as follows, please see line 368.

The formula for determining the relative water content of leaves W(%) was: (m1-m3)/(m2-m3)×100%.

Comment 12: Discussion: It would be better to add subheadings around a specific and discussed topic.

Response 12: Gratitude for your suggestions. The discussion sections have been sub-titled according to the focus of each section, respectively, " Photosynthetic performance in drought and re-watering of WDR ", " Effects of antioxidant enzyme activities on photosynthetic production in drought and re-watering of WDR ", and " Effects of osmoregulatory substances on photosynthetic production in drought and re-watering of WDR ".

Comment 13: There are some grammatical and spelling mistakes throughout the manuscript. Therefore, the manuscript should be polished by native English speakers.

Response 13: The advice is appreciated. The revised manuscript was polished by a English native speaker.

Reviewer 2 Report

This paper is largely concerned with physiological features of drought tolerance in rice, with no molecular or related study done to offer proof of concept for the results to be published in International journal of Molecular Sciences. As a result, this work is unsuitable for publication in this journal (International Journal of Molecular sciences). I would advise that the editor move this paper to Stresses (MDPI journal) or another journal dealing with abiotic stress.

Author Response

Response to Reviewer #2

Comment 1: The introduction provide sufficient must be improved.

Response 1: Much obliged to you for your suggestion. In the introduction, the performance of antioxidant enzymes and osmoregulatory substances in drought and rehydration is described more than in photosynthetic systems. This issue was identified and now relevant literature studies have been added. Please see lines 58 to 66.

Drought decreases photosynthesis and reduces the rate of electron transfer and the maximum quantum production of the crop, while different degrees and durations of drought cause different degrees of damage. All photosynthetic functions, especially the PSII system, can be recovered in sweet sor-ghum after drought rehydration, while the photosynthetic rate and photosynthetic electron transfer capacity of maize cannot be restored to pre-drought conditions under prolonged drought[14–16]. In addition, according to Zegada-Lizarazu&Monti (2013), photosystem II (PSII) photochemistry has an effective self-regulatory function for sweet sorghum, contributing to its high drought tolerance and photosynthetic resilience[14]

Comment 2: All the cited references are relevant to the research.

Response 2: Thank you very much for your positive comment.

Comment 3: The research design is appropriate.

Response 3: Your recognition is much appreciated.

Comment 4: The methods are adequately described.

Response 4: Much thanks for your approval.

Comment 5: The results are clearly presented.

Response 5: For your recognition, we are grateful.

Comment 6: The conclusions supported by the results must be improved.

Response 6: With thanks for your suggestion, the conclusion has now been revised as follows. Please see lines 462 to 471.

  1. i) Drought stress decreases photosynthetic rates and generally increase the activi-ties of antioxidant enzymes and the content of osmoregulatory substances in both rice cultivars. WDR (HY73) can be rapidly recovered to its initial pre-drought state after rehydration, while photosynthetic potential is unable to recover after rehydration for drought sensitive cultivar (HHZ).
  2. ii) High photosynthetic potentials are mainly relying on high POD activity to eliminate reactive oxygen species for protection of cell membrane integrity in WDR. Also, high SP content in WDR has positive role to maintain a steady intracellular os-motic pressure. The both factors could be essential physiological bases for the rapid recovery of photosynthetic potential after rehydration in WDR.

Comment 7: This paper is largely concerned with physiological features of drought tolerance in rice, with no molecular or related study done to offer proof of concept for the results to be published in International journal of Molecular Sciences. As a result, this work is unsuitable for publication in this journal (International Journal of Molecular sciences). I would advise that the editor move this paper to Stresses (MDPI journal) or another journal dealing with abiotic stress.

Response 7: It is highly appreciated for your comments and positive feedback. Following careful study of your views, we think this article could be a suitable one for publication in International journal of Molecular Sciences for the following reasons.

  • The paper analyzes the physiological characteristics and photosynthetic physiological changes of water-saving and drought-resistant rice after drought and rehydration to determine its photosynthetic adaptation mechanism, which is in line with the theme of the special issue Assimilate Production and Allocation in Plants under Abiotic Stress.
  • In addition, there are severalrecent articles without molecular or related studies as follows, 
  1. Using Exogenous Melatonin, Glutathione, Proline, and Glycine Betaine Treatments to Combat Abiotic Stresses in Crops(https://doi.org/10.3390/ ijms232112913), published in a special issue of International Journal of Molecular Sciences, Physiological and Molecular Studies on Plant Bioactive Compounds under Environmental Stresses.
  2. Mineral Monitorization in Different Tissues of Solanum tuberosum L. during Calcium Biofortification Process (https://doi.org/10.3390/horticulturae8111020 (registering DOI)), belongs to the section Plant Nutrition.
  • Study on Chromium Uptake and Transfer of Different Maize Varieties in Chromium-Polluted Farmland (https://doi.org/10.3390/su142114311(register ing DOI)), published in a special issue of Remediation of Contaminated Soil and Wastewater Treatment.
  1. The Growth and Physiological Characteristics of the Endangered CAM Plant, Nadopungnan (Sedirea japonica), under Drought and Climate Change Scenarios (https://doi.org/10.3390/f13111823(registering DOI)), belongs to the section Forest Ecophysiology and Biology.

Thanks again for your comments, we have revised them accordingly and hope that they would meet with your approval. We sincerely hope that this revised manuscript would address all your comments and suggestions.

Reviewer 3 Report

Water-saving and drought-resistant rice (WDR) has high yield potential in drought. But little is known about the physiological adaptation mechanisms of drought and rehydration in WDR. In this study, WDR cultivar HY73 and drought-sensitive cultivar HLY898 were selected to reveal the dynamic changes of physiological characteristics when response to drought and rewatering. Authors supposed that high yield is related to improving self-protection abilities and adjustment abilities from antioxidant enzymes and osmoregulatory substances perspectives, respectively, and then ensuring high photosynthetic rate during drought and re-watering. In general, this study would provide some useful conclusions for comprehending the mechanisms of drought-resistance and high yield of WDR. However, following questions need to be addressed before considering this MS for publication in IJMS. 

(1) Title: it would be better to wrote “Physiological Adaptation Mechanisms of Drought and Rewatering in Water-saving and Drought-resistant Rice”, because not only “photosynthetic adaptation mechanisms” was involved in this study.

(2) Abstract: lines 24 to 27, SOD, POD, CAT, SP, MDA need to be defined where they were firstly presented.

(3) Introduction: antioxidant enzymes and osmoregulatory substances were emphatically introduced in drought. However, photosynthetic adaptation characteristics are clearly inadequate in drought and re-watering, especially for rice plants.

(4) Results: What are the (-) (+) meaning in 2.3 and 2.4? It is difficult to understand the results. In addition, the biological repeats should be referenced in figure captions.

(5) Figs 6 and 7. Amax is usually represented maximum net photosynthetic rate instead of Pmax. Please check the issues throughout the text.

(6) Discussion: Lines 298 to 300, “Therefore, we venture to speculate that HY73 uses Pro to maintain osmotic pressure homeostasis during drought and SP as a supplement to cellular water loss”. I think the conclusion is not rigorous. It might be one of the reasons.

(7) Methods: Line 354, why the maximal PAR was set as 1500 μmol·m-2· s-1 instead of 1800 μmol·m-2· s-1 or higher PAR. Please explain it in text.

(8) 4.3 part. The multiple comparisons between treatments were not clearly described. Please re-write it.

(9) I suggest working with a native English speaker to polish the manuscript.

Author Response

Response to Reviewer #3

Comment 1: Title: it would be better to wrote “Physiological Adaptation Mechanisms of Drought and Rewatering in Water-saving and Drought-resistant Rice”, because not only “photosynthetic adaptation mechanisms” was involved in this study. 

Response 1: With many thanks, we gladly accept your suggestion. Admittedly parts of this title were not evident in the text, and we have now rewritten the title: Physiological Adaptation Mechanisms to Drought and Rewatering in Water-saving and Drought-resistant Rice.

Comment 2: Abstract: lines 24 to 27, SOD, POD, CAT, SP, MDA need to be defined where they were firstly presented.

Response 2: Thank you very much. Abbreviations appearing for the first time in the text have been re-added in their full spelling form: superoxide dismutase (SOD), peroxidase (POD), catalase (CAT), ascorbate peroxidase (APX), malondialdehyde (MDA), soluble protein (SP).

Comment 3: Introduction: antioxidant enzymes and osmoregulatory substances were emphatically introduced in drought. However, photosynthetic adaptation characteristics are clearly inadequate in drought and re-watering, especially for rice plants.

Response 3: Many thanks for your advice. The following additions have been made to the literatures on photosynthetic adaptation characteristics after drought and rehydration. Please see lines 59 to 66.

Drought decreases photosynthesis and reduces the rate of electron transfer and the maximum quantum production of the crop, while different degrees and durations of drought cause different degrees of damage. All photosynthetic functions, especially the PSII system, can be recovered in sweet sorghum after drought rehydration, while the photosynthetic rate and photosynthetic electron transfer capacity of maize cannot be restored to pre-drought conditions under prolonged drought[14–16]. In addition, according to Zegada-Lizarazu&Monti (2013), photosystem II (PSII) photochemistry has an effective self-regulatory function for sweet sorghum, contributing to its high drought tolerance and photosynthetic resilience[14].

Comment 4: Results: What are the (-) (+) meaning in 2.3 and 2.4? It is difficult to understand the results. In addition, the biological repeats should be referenced in figure captions.

Response 4: Your guidance is much appreciated. The meaning of the expressions (-) and (+) in 2.3 and 2.4 is not really clear, as they represent a decrease and an increase respectively. They have now been replaced in the text with words that match their meaning, with lines 186 to 189 and lines 203 to 211being rewritten in the following form:

Comment 5: Figs 6 and 7. Amax is usually represented maximum net photosynthetic rate instead of Pmax. Please check the issues throughout the text.

Response 5: Very grateful for the prompt, the full text has now been checked and replaced with Amax instead of Pmax.

Comment 6: Discussion: Lines 298 to 300, “Therefore, we venture to speculate that HY73 uses Pro to maintain osmotic pressure homeostasis during drought and SP as a supplement to cellular water loss”. I think the conclusion is not rigorous. It might be one of the reasons.

Response 6: Comments are appreciated. This sentence does not seem convincing enough as a conclusion and lines 296 to 300 has now been re-edited to change the sentence as follows, please see lines 329 to 333.

SP is the main product of plant photosynthesis and a nonstructural carbohydrate [79]. Changes in SP content under drought stress can not only reveal its osmoregulatory capacity but also be used to probe the photosynthetic assimilation capacity of the plant itself. The correlations between Amax of HY73 and SP and Pro showed differences, in a sense verifying this point.

Comment 7: Methods: Line 354, why the maximal PAR was set as 1500 μmol·m-2· s-1 instead of 1800 μmol·m-2· s-1 or higher PAR. Please explain it in text..

Response 7: Gratitude for your comments. The maximal PAR setting in the experiment was determined based on the range of light intensities during normal daylight from 09:00-12:00 at the test site last year, for the purpose of rapid light acclimatization of the leaves and to avoid phenomena such as photoinhibition.

Comment 8: 4.3 part. The multiple comparisons between treatments were not clearly described. Please re-write it.

Response 8: Greatly appreciated, part 4.3 has been rewritten. More details were introduced in the part.

Comment 9: I suggest working with a native English speaker to polish the manuscript.

Response 9: The advice is much appreciated. The revised manuscript was polished by a English native speaker.